# The Genomic, Transcriptomic, and Immunologic Landscape of *HRAS* Mutations in Solid Tumors

**DOI:** 10.3390/cancers16081572

**Published:** 2024-04-19

**Authors:** Samuel A. Kareff, Asaad Trabolsi, Harris B. Krause, Timothy Samec, Andrew Elliott, Estelamari Rodriguez, Coral Olazagasti, Dionysios C. Watson, Matias A. Bustos, Dave S. B. Hoon, Stephanie L. Graff, Emmanuel S. Antonarakis, Sanjay Goel, George Sledge, Gilberto Lopes

**Affiliations:** 1Department of Graduate Medical Education, University of Miami Sylvester Comprehensive Cancer Center/Jackson Memorial Hospital, Miami, FL 33136, USAaxt312@med.miami.edu (A.T.); 2Caris Life Sciences, Phoenix, AZ 85040, USA; hkrause@carisls.com (H.B.K.);; 3Division of Medical Oncology, Department of Medicine, University of Miami Sylvester Comprehensive Cancer Center, Miami, FL 33136, USA; estelarodriguez@miami.edu (E.R.);; 4Division of Translational Molecular Medicine, St. Johns’ Cancer Institute at Providence Saint John’s Health Center, Santa Monica, CA 90404, USA; bustosm@jwci.org (M.A.B.); dave.hoon@providence.org (D.S.B.H.); 5Department of Medicine, Lifespan Cancer Institute, Providence, RI 02903, USA; 6Division of Hematology, Oncology, and Transplantation, University of Minnesota Masonic Cancer Center, Minneapolis, MN 55455, USA; anton401@umn.edu; 7Division of Medical Oncology, Rutgers University, New Brunswick, NJ 08901, USA; sanjay.goel@rutgers.edu

**Keywords:** HRAS, tipifarnib, translational oncology, precision oncology, biomarkers

## Abstract

**Simple Summary:**

In the era of precision medicine, translational oncology seeks to identify new, targeted therapies for tumors with rare genetic mutations. Patients receiving targeted therapy are known to have better outcomes (i.e., live longer and better) compared to those receiving non-targeted therapies. In this analysis, we retrospectively analyzed 63,873 tumor tissues to better understand the rare *HRAS* mutation. We found that only 0.8% of tumors are *HRAS* mutant, and these tumors look different at the molecular level and behave differently on a clinical level. Targeted therapies for patients with *HRAS* mutations, such as tipifarnib, currently exist and are being tested in various tumor types. Our study seeks to add to the limited information currently known about this rare genetic mutation.

**Abstract:**

Tipifarnib is the only targeted therapy breakthrough for *HRAS*-mutant (*HRAS*mt) recurrent or metastatic head and neck squamous cell carcinoma (HNSCC). The molecular profiles of *HRAS*mt cancers are difficult to explore given the low frequency of *HRAS*mt. This study aims to understand the molecular co-alterations, immune profiles, and clinical outcomes of 524 *HRAS*mt solid tumors including urothelial carcinoma (UC), breast cancer (BC), non-small-cell lung cancer (NSCLC), melanoma, and HNSCC. *HRAS*mt was most common in UC (3.0%), followed by HNSCC (2.82%), melanoma (1.05%), BC (0.45%), and NSCLC (0.44%). *HRAS*mt was absent in Her2+ BC regardless of hormone receptor status. *HRAS*mt was more frequently associated with squamous compared to non-squamous NSCLC (60% vs. 40% in *HRAS*wt, *p* = 0.002). The tumor microenvironment (TME) of *HRAS*mt demonstrated increased M1 macrophages in triple-negative BC (TNBC), HNSCC, squamous NSCLC, and UC; increased M2 macrophages in TNBC; and increased CD8+ T-cells in HNSCC (all *p* < 0.05). Finally, *HRAS*mt was associated with shorter overall survival in HNSCC (HR: 1.564, CI: 1.16–2.11, *p* = 0.003) but not in the other cancer types examined. In conclusion, this study provides new insights into the unique molecular profiles of *HRAS*mt tumors that may help to identify new targets and guide future clinical trial design.

## 1. Introduction

The *KRAS*, *NRAS,* and *HRAS* genes comprise the *RAS* family in humans, which encode four similar proteins: *KRAS*4A/B, *NRAS*, and *HRAS* [1]. RAS proteins regulate tumorigenesis [2] and, when activated, drive cellular processes such as proliferation, differentiation, cell adhesion, and cell migration via the mitogen-activated protein (MAP) kinase pathway [3]. RAS proteins are among the most frequently mutated in solid tumors [1], and targeted therapies against the activating mutation *KRAS* G12C are clinically active for various solid tumors including non-small-cell lung (sotorasib [4] and adagrasib [5]) and pancreatic (sotorasib [6]) cancers.

*HRAS* (*Harvey rat sarcoma viral oncogene homolog*) is a less commonly mutated RAS family member whose membrane localization and signaling are dependent on the post-translational addition of a farnesyl lipid moiety (farnesylation) [7,8]. The frequency of somatic *HRAS* mutations in solid tumors has been estimated to be between 1.0 and 6.9% [9,10] depending on the cohort of patients studied. Tipifarnib, a farnesyltransferase inhibitor (FTI) affecting the post-translational modification of *HRAS* to prevent membrane binding, demonstrated clinical activity in an open-label Phase-II study of patients with *HRAS*-mutated (*HRAS*mt) head and neck cancer, with an objective response rate of 55% [11]. This agent later gained breakthrough therapy designation by the United States Food and Drug Administration (FDA) in February 2021 for patients with head and neck squamous cell carcinoma with a tumor variant allele frequency (VAF) > 20% after progression on platinum-based chemotherapy based on the clinical trial NCT02383927 [11]. As of August 2023, there are at least four active clinical trials evaluating tipifarnib in other tumor types, including non-small-cell lung cancer (NSCLC) (NCT03496766), lymphoma or histiocytic disorders (NCT04284774), urothelial carcinoma (NCT02535650), and advanced solid malignancies (NCT04284774 and NCT04865159).

Given the prevalence of *HRAS*mt in other solid tumors, we sought to investigate the molecular characteristics and clinical outcomes of *HRAS*mt in five solid tumor types to determine the prevalence of *HRAS*mt, interrogate molecular co-alterations, and explore the potential role of *HRAS*mt as a prognostic or therapeutic biomarker.

## 2. Materials and Methods

### 2.1. Cohort Information and Tumor Types

Urothelial carcinoma (N = 4605, UC), breast cancer (N = 15,834, BC), non-small-cell lung cancer (N = 34,310, NSCLC), melanoma (N = 5217), and head and neck squamous cell carcinoma (N = 3907, HNSCC) tumors that underwent comprehensive at Caris Life Sciences (Phoenix, AZ, USA) were included in this study. BC tumors were divided based on receptor subtypes (i.e., hormone receptor [HR]+, Her2+, and triple-negative breast cancer [TNBC]). NSCLC tumors were divided into adenocarcinoma, squamous cell carcinoma, and others based on histology.

### 2.2. Next-Generation Sequencing-592 Gene Panel (NGS-592)/Whole Exome Sequencing (WES)

NGS-592 or whole exome sequencing (WES) was performed for 191,767 solid tumors sequenced at Caris Life Sciences. These assays and their analyses were recently reported [12].

WES was performed on genomic DNA isolated from a micro-dissected, FFPE tumor sample using the Illumina NovaSeq 6000 sequencers (Illumina, Inc.; San Diego, CA, USA). A hybrid pull-down panel of baits designed to enrich for 700 clinically relevant genes at high coverage and high read-depths was used, along with another panel designed to enrich for an additional >20,000 genes at a lower depth. The performance of the WES assay was validated for sequencing variants, copy number alteration, tumor mutational burden (TMB), and microsatellite instability (MSI). The WES assay was validated to 50 ng of input and had a positive predictive value of 0.99 against a previously validated NGS assay. WES can detect variants in samples with tumor nuclei as low as 20% and detects down to a 5% variant frequency, with an average depth of at least 500×.

### 2.3. Identification of Genetic Variants

The genetic variants identified were interpreted by board-certified molecular geneticists and categorized as ‘pathogenic’, ‘likely pathogenic’, ‘variant of unknown significance’, ‘likely benign’, or ‘benign’, according to the American College of Medical Genetics and Genomics (ACMG) standards. When assessing mutation frequencies of individual genes, ’pathogenic’, and ‘likely pathogenic’ were counted as mutations (mt), while ‘benign’ and ‘likely benign’ variants and ‘variants of unknown significance’ (VUS) were excluded. *HRAS* pathogenic mutants were further divided based on the affected amino acid (Q61, G12, G13, and miscellaneous [for all the mutations that were not present in Q61, G12, or G13]) (Table A1).

### 2.4. Whole Transcriptome Sequencing

FFPE specimens underwent pathology review to diagnose the percentage tumor content and tumor size. A minimum of 10% of tumor content in the area for microdissection was required to enable the enrichment and extraction of tumor-specific RNA. A Qiagen RNA FFPE tissue extraction kit was used, and the RNA quality and quantity were determined using the Agilent TapeStation. Biotinylated RNA baits were hybridized to the synthesized and purified cDNA targets, and the bait–target complexes were amplified in a post-capture PCR reaction. The resultant libraries were quantified and normalized, and the pooled libraries were denatured, diluted, and sequenced. Transcriptions per million molecules (TPM) were generated using the Salmon expression pipeline for transcript counting. 

### 2.5. Immune Signatures

Immune cell fraction was calculated via the deconvolution of WTS data using quanTIseq. QuanTIseq is an immune deconvolution algorithm that utilizes RNA transcripts known to be expressed in specific immune cell types to deconvolute bulk RNA sequencing data and predict the different immune cell fraction present in the bulk RNA sequencing data [13]. WTS data were also used to calculate a T-cell-inflamed score, as previously described [14].

### 2.6. Microsatellite Insability/Mismatch Repair Deficiency (MSI-h/MMR) Status

A combination of multiple test platforms was used to determine MSI-H or dMMR status of the tumors profiled, including fragment analysis (FA; Promega, Madison, WI, USA), immunohistochemistry (IHC; MLH1, M1 antibody; MSH2, G2191129 antibody; MSH6, 44 antibody; and PMS2, EPR3947 antibody [Ventana Medical Systems, Inc., Tucson, AZ, USA]), and NGS (>2800 target microsatellite loci were examined and compared to the reference genome hg19 from the University of California). The two platforms generated highly concordant results, as previously reported, and in rare cases of discordant results, the MSI-H or MMR status of the tumor was determined in the order of IHC and NGS.

### 2.7. Tumor Mutational Burden (TMB)

TMB was measured by counting all non-synonymous missense, nonsense, inframe insertion/deletion, and frameshift mutations found per tumor that had not been previously described as germline alterations in dbSNP151 or the Genome Aggregation Database (gnomAD) or benign variants identified by Caris’ geneticists. A cutoff point of ≥10 mutations (mt) per MB was used based on the KEYNOTE-158 pembrolizumab trial, which showed that patients with a TMB of ≥10 mt/MB (TMB-H) across several tumor types had higher response rates than patients with a TMB of <10 mt/MB [15,16]. 

### 2.8. Immunohistochemistry (IHC)

IHC was performed on FFPE sections of glass slides. Slides were stained using automated staining techniques, per the manufacturer’s instructions, and optimized and validated per CLIA/CAP and ISO requirements. Staining was scored for intensity (0 = no staining; 1+ = weak staining; 2+ = moderate staining; 3+ = strong staining) and percentage (0–100%). PD-L1 (SP142) positive (+) staining was defined as ≥2+ and ≥5%, and 22c3 (+) was defined as TPS ≥1%. ER or PR + was defined as ≥1+ and ≥1%. HER2/Neu + was defined as ≥3+ and >10%.

### 2.9. Clinical Outcomes

Real-world overall survival was obtained from insurance claims and calculated from tissue collection to last contact. This surrogate outcome has been utilized previously [17]. Kaplan–Meier estimates were calculated for molecularly defined patients.

### 2.10. Statistics and Reproducibility

Descriptive analyses were conducted utilizing the Mann–Whitney U (scipy V.1.9.3) and X^2^/Fisher Exact tests (R v.3.6.1) for continuous and categorical variables, respectively. *p*-values were adjusted for multiple comparisons, with *p* < 0.05 considered significant.

## 3. Results

### 3.1. Prevlance of HRASmt and Characteristics of Cohort 

The Caris Life Sciences biobank was queried for the frequency of HRASmt. Five tumor types with the highest number of HRASmt cases were chosen for a deeper analysis of their molecular and immunologic landscapes. A total of 63,873 tumor tissues were analyzed, and among the entire cohort, 0.82% of tumors were HRASmt. HRASmt (total N = 524) accounted for 3.0% of urothelial carcinomas (UC, N = 4605), 0.50% of breast cancer cases (BC, N = 15,843), 0.44% of non-small-cell lung cancer cases (NSCLC, N = 34,310), 1.05% of melanomas (N = 5217), and 2.82% of HNSCCs (N = 3907) (Table 1). The median age in years of patients at the time of specimen collection was 72 in UC, 60 in BC, 69 in NSCLC, 67 in melanoma, and 64 in HNSCC. Aside from BC, only HNSCC demonstrated significantly greater HRASmt prevalence among females (37.3% vs. 23.0%, *p* = 0.001). Among BC subtypes, HRASmt cancers were exclusively observed in hormone receptor-positive (HR+)/Her2- (17/8057, 0.21%,) and triple-negative (TNBC) (54/4457, 1.21%) cases. HRASmt were absent in Her2+ BC, irrespective of HR expression (0/1210, 0%). A significantly higher proportion of HRASmt cancers were squamous NSCLC (90/150, 60% total) versus non-squamous NSCLC (60/150, 40%, *p* = 0.002).

The distribution of *HRAS* codon mutations varied by cancer type, with a higher prevalence of *HRAS* G13 mutations in HNSCC (42/112, 37.5%), squamous NSCLC (41/90, 45.6%), and melanoma (22/55, 40%) and a higher prevalence of Q61 mutations in non-squamous NSCLC (18/34, 52.9%) compared to other mutations (Figure 1A). *HRAS* expression was significantly higher in HRASmt compared to HRASwt tumors (transcripts per parts million [TPM]) among all investigated tumor types (Figure 1B). Notably, there was minimal variation (not significant, *p* > 0.05) in *HRAS* expression across the investigated mutation types, except for UC, in which the G12 mutation had significantly higher expression compared to *HRAS* Q61 and G13 mutations (*p* < 0.05, Figure 1C).

### 3.2. HRASmt Co-Alterations and Biomarkers

TP53 mutations were significantly more prevalent in HRASwt for both TNBC and UC. RB1 mutations were more common in HRASwt UC. PIK3CA and PIK3R1 mutations were more prevalent in HRASmt TNBC only. RAF1 and BRAF mutations had a higher prevalence in HRASmt melanoma and UC, respectively, whereas *NRAS* mutations were more prevalent in HRASwt melanoma. Increased copy number alterations of FGF19, FGF3, FGF4, and CCND1 (all located on 11q13 amplicon) were observed in HRASwt HNSCC. HNSCC HRASmt had an increased prevalence of the TERT promoter, FAT1, NOTCH1, CASP8, and CTCF mutations (Figure 2, *p* < 0.05 for all). The prevalence of *KRAS* mutation was not significantly different between HRASmt and HRASwt tumors across all investigated tumor types. However, significant differences in the prevalence of *KRAS*mt G12C were observed in HRASmt vs. HRASwt in HR+/HER2- BC (5.9 vs. 0.1%), HNSCC (2.7 vs. 0.1%), squamous NSCLC (8.9 vs. 2.6%), and UC (5.8 vs. 0.4%) (all *p* < 0.05).

No significant difference in the prevalence of PD-L1-positive tumors was observed via IHC between HRASmt and HRASwt tumors (Figure 3A, *p* > 0.05). Additionally, there was no significant difference in the prevalence of mismatch repair deficiency/high microsatellite instability (dMMR/MSI-H) (Figure 3B, *p* > 0.05) or high tumor mutational burden (TMB-H) (Figure 3C, *p* > 0.05) among HRASmt compared to HRASwt tumors.

### 3.3. Tumor Microenvironment (TME) of HRASmt

No statistically significant difference in the prevalence of T-cell-inflamed tumors between HRASmt and HRASwt tumors was observed (Figure 4A, *p* > 0.05). TNBC, HNSCC, and UC tumors had distinct tumor microenvironments (TMEs) between HRASmt and HRASwt tumors. Using quanTIseq immune deconvolution to infer the prevalence of immune infiltrates, M1 macrophage immune infiltrates were significantly higher in HRASmt versus HRASwt tumors for TNBC, HNSCC, squamous NSCLC, and UC. M2 macrophage immune infiltrates were more prevalent in HRASmt TNBC tumors but were less prevalent in HNSCC and UC tumors. Neutrophil infiltrates were more prevalent in HRASmt TNBC, HNSCC, and UC tumors. Conversely, in the TME of HRASwt TNBC, HNSCC, and UC, dendritic cell infiltrates, B cells, and NK cells were more prevalent. Finally, CD8+ T-cell infiltrates were more prevalent in HRASmt HNSCC but less common in HRASmt UC (*p* < 0.05 for all described, Figure 4B,C).

### 3.4. Clinical Outcomes

Overall survival (OS) was significantly shorter for HRASmt versus HRASwt HNSCC (HR: 1.564, CI: 1.16–2.11, *p* = 0.003) (Figure 5B). Shorter OS in HRASmt was not associated with any specific *HRAS* codon mutations in HNSCC (Figure A1). Shorter OS in HRASmt was demonstrated regardless of exposure to cetuximab (no cetuximab HR: 1.515; cetuximab HR: 2.084) or immune checkpoint inhibitors (ICI) (no ICI HR 1.455; ICI HR 1.961) (all *p* < 0.05) (Figure A2). Worse outcomes were also observed with HRASmt HNSCC treated with chemotherapy (HR 1.909, *p* = 0.004); this trend was not seen for samples without chemotherapy exposure (HR 1.407, *p* = 0.097) (Figure A2). No significant differences in OS were seen in the other tumor types studied when comparing HRASmt with HRASwt (Figure 5A, Table 2, and Figure A1).

## 4. Discussion

We highlighted the prevalence, characteristics, and outcomes of *HRAS*mt tumors. Among the entire cohort, 0.82% of tumors were *HRAS*mt. This estimate is slightly lower than previously described [9]. However, the disease-specific prevalences reported in this study of 3.0% in UC, 2.82% in HNSCC, and 1.05% in melanoma mirror the literature [18,19,20,21].

Various *HRAS* codon mutations have been investigated for molecular targeting. For example, *HRAS* Q61L has been considered as a possible target in NSCLC [22] given the particularly poor prognosis and clinical features described in the literature. This target has also been previously described as a candidate for treatment in rare BC histologies [23]. Within all BCs, *HRAS*mt BCs were mutually exclusive with the Her2+ subtype. This is in line with the biology of Her2+ BC, which is primarily driven by the activation of c-Src and not the MAP kinase pathway [24]. However, Her2+ BC may be under-represented in this dataset. Our analysis included 8.8% Her2+ BC, lower than the 15–20% prevalence of Her2+ BC typically reported. This discrepancy may reflect patterns of genomic profiling utilization in clinical practice [25]. Conversely, rarer BC subtypes, such as malignant breast adenomyoepithelioma, are known to be driven by the MAP kinase pathway alterations, such as *HRAS* Q61 recurrent hotspot mutations [26]. Next, there was a statistically higher prevalence of *HRAS*mt in squamous compared to non-squamous NSCLC. In contrast to our cohort, previous studies suggested that *HRAS*mt was more prevalent in non-squamous (i.e., adenocarcinoma) NSCLC, likely due to a lower sample size in this prior publication (*N* = 39 vs. the current study, *N* = 34,310 [27]. Given these findings, *HRAS*mt may represent a key therapeutic target in HR+ BC, TNBC, and squamous NSCLC.

There was a significant association between *TP53*mt and *HRAS*wt in TNBC and UC. In line with this observation, a tendency towards mutual exclusivity between *TP53*mt and *HRAS*mt in UC has been previously reported in the literature [28]. With inhibitors for *TP53*wt *HRAS*mt UC under development [29], our results provide further evidence for targeting *HRAS*mt UC. Finally, the alterations observed in our HNSCC cohort echo a previously reported dataset in which *CASP8*, *TERT*, and *NOTCH1* were also frequent co-mutations in *HRAS*mt HNSCC, while *CCND1* had higher amplification in *HRAS*wt compared to *HRAS*mt [20] (Figure 2). However, we also report a significantly increased co-occurrence of *HRAS*mt with *CTCF* and *FAT1* mutations (Figure 2). Finally, past work has suggested that RAS family mutations are mutually exclusive [30] or preferentially enriched for concomitant downstream RAS mutations (e.g., BRAF) [31]. However, the mutual exclusivity of RAS family mutations has come into question, especially given the prevalence of concomitant mutations in solid tumors such as colorectal cancer [32]. Our results confirm the presence of multiple isoforms of RAS mutations (i.e., *KRAS*mt) within our cohort of *HRAS*mt solid tumors, thereby refuting previous observations that RAS family mutations are mutually exclusive in solid tumors such as urothelial carcinoma [33].

While smaller cohorts of other tumors, such as medullary thyroid carcinoma [34], have reported significant associations with PD-L1 positivity and TMB and *HRAS*mt status, our cohort clarifies that there was no statistical relationship demonstrated between these biomarkers and the investigated cancer types according to *HRAS*mt status. Additionally, a relatively immunologically “cold” TME in *HRAS*mt TNBC and UC was observed, as demonstrated by decreased CD4+ and CD8+ T cells in each cancer type, although none of these differences reached statistical significance. This observation suggests that certain ICIs, such as monotherapy with PD-(L)1 inhibitors, are unlikely to play a role in these *HRAS*mt tumors. This is in contrast to a small cohort of *HRAS*mt HCSCC that noted an increase in CD8+ T cells within the tumor microenvironment (TME), analyzed using the ESTIMATE immune score [35]. However, it is worth noting that the small cohort of *HRAS*mt HNSCCs reporting these TME discrepancies with our study was treated with immune checkpoint inhibitors, including anti-PD-(L)1 and -CTLA-4 agents, prior to data acquisition [35,36].

*HRAS*mt status was associated with poorer clinical outcomes in HNSCC, and no difference in OS was observed between the different *HRAS* codon mutations. Similar findings were recently reported for 249 *HRAS*mt HNSCC samples with median disease-free survival (DFS) of 4.0 months and OS between 15 and 25.5 months [20], which were slightly longer than those observed in our cohort. This confirms the aggressive clinical nature of *HRAS*mt HNSCC, as has been observed previously [37]. Moreover, there were poorer OS outcomes demonstrated for *HRAS*mt HNSCC with (mOS 9.9 months, *p* = 0.028) or without (mOS 11.5 months, *p* = 0.015) exposure to cetuximab. This component of the EXTREME regimen [38] is typically reserved for second-line therapies and beyond and may be associated with other factors secondary to this subset of aggressive disease biology. A similar trend was seen with exposure to ICIs, again possibly reflecting acquired disease resistance.

Our findings support several therapeutic hypotheses and approaches. Tipifarnib initially demonstrated clinical activity in recurrent, metastatic *HRAS*mt salivary gland cancer, paving the way for accelerated FDA approval for FTI [37]. Based on this success, FTIs have been touted as novel therapeutic approaches in other *HRAS*mt cancers, such as rhabdomyosarcoma [39], UC [40], anaplastic [41] and dedifferentiated thyroid cancers [42], and salivary duct carcinoma [43]. It is important to note that FTIs act on farnesyl transferase (FT) and, therefore, have the potential for off-target toxicity, whereas other RAS-targeting inhibitors (e.g., sotorasib and adagrasib) bind directly to the mutated protein without similar side effects. Sotorasib and adagrasib have similar mechanisms of action through covalent binding of the cysteine 12 site within the *KRAS* G12C protein, rendering *KRAS* inactive and preventing cell proliferation, effectively halting cancer cell progression [44,45]. Because of the specific G12C biding activity, there is a marked reduction in off-target binding both in vitro and in vivo [45,46,47,48,49]. FTIs, however, may cause the displacement of *HRAS*wt from cell membranes via the inhibition of FT, causing the destabilization of non-malignant cells [50]. Additionally, these more targeted agents have demonstrated clinical activity across a broader range of tumors compared to FTIs. In addition to the aforementioned studies in NSCLC (NCT03496766), lymphoma or histiocytic disorders (NCT04284774), UC (NCT02535650), and advanced solid malignancies (NCT04284774 and NCT04865159), our observations may guide additional investigation into the use of FTIs in *HRAS*mt TNBC, HR+ BC, and squamous NSCLC.

Additional therapeutic targets and pathways might also be studied. *HRAS*mt cancers have been posited as novel targets in vitro for MEK and mTOR inhibitors. The murine Ba/F3 cell line has demonstrated the increased sensitivity of both classes of drugs towards the *HRAS* isoforms Q16L, Q16R, and G12V [51]. Murine models have demonstrated activity of MEK inhibition alone in other *HRAS*mt diseases, such as Costello syndrome [52]. Finally, mTOR inhibitors combined with ERK inhibition demonstrated activity against *HRAS*mt G12V-driven autochthonous sarcoma [53]. Given the current approval of everolimus in the advanced/metastatic HR+ BC setting [54], future analyses may consider determining if *HRAS*mt cancers serve as a predictor of response.

The limitations of our study include its retrospective analysis and use of surrogate measures (e.g., insurance claims) for outcomes. Furthermore, the five tumor types represented here may not represent the histologies with the highest absolute percentages of *HRAS*mt. For example, pheochromocytomas demonstrate *HRAS*mt at levels as high as 12.35% in publicly available datasets [55], though the absolute number of cases in our biobank limited this tumor’s inclusion. Moreover, our cohort lacks racial and ethnic data. *HRAS*mt cancers are known to be more prevalent in Hispanic White and African American patients with HNSCC [56], for example, and our cohort cannot speculate on the racial or ethnic characteristics of the samples analyzed. Finally, the dataset utilized may under-represent Her2+ BC, as previously mentioned. This limitation may be ameliorated with future analyses given that standard-of-care molecular testing has now widely entered oncology practice. Future studies ought to consider prospective design, include demographic data, and propose solutions to ameliorate barriers to molecular testing.

## 5. Conclusions

To the best of our knowledge, our findings represent the largest cohort to date reporting on the genomic, transcriptomic, and immunologic landscapes of *HRA*Smt solid tumors. The unique genomic and immunologic profiles of *HRAS*mt tumors may guide researchers in identifying and trialing new targeted agents in this subset of molecularly driven cancers.

## Figures and Tables

**Figure 1 cancers-16-01572-f001:**
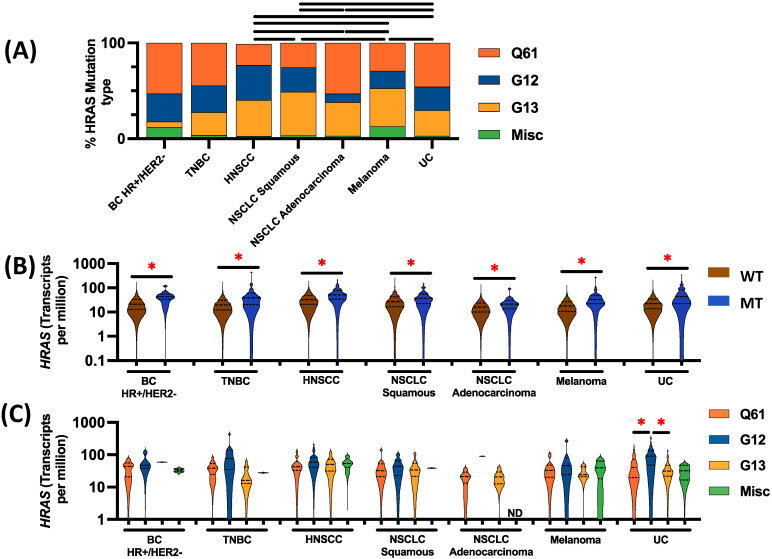
(**A**) Frequency of different *HRAS* mutations across investigated solid tumors. Q61: 182A>G, 182A>T, 181C>A, and 180-182del_insTCT. G12: 34G>A, 34G>T, 35G>A, 35G>T, and 35G>C. G13: 37G>C, 38G>T, and 38G>A. Misc is defined as all other pathogenic *HRAS* mutations. (**B**) *HRAS* expression between *HRAS* mutant (MT) and wild-type (WT) tumors. (**C**) *HRAS* expression across different *HRAS* mutations (red asterisk [*] or black bars indicate *p* < 0.05).

**Figure 2 cancers-16-01572-f002:**
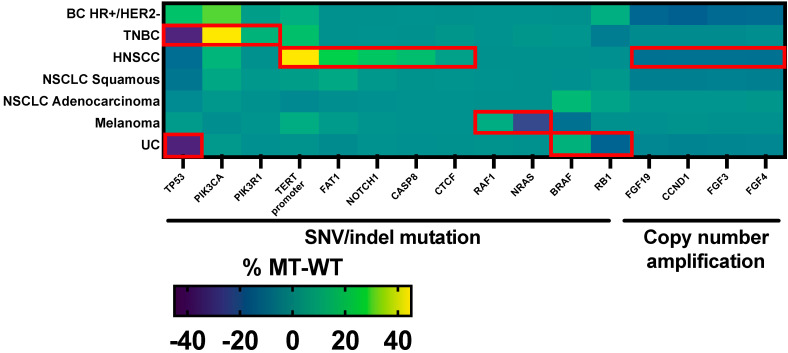
Prevalence of genomic alteration in *HRAS* MT–WT tumors (genes shown had a statistically significant difference in mutation prevalence for at least one of the investigated cancer types). Red box indicates statistical significance (*p* < 0.05).

**Figure 3 cancers-16-01572-f003:**
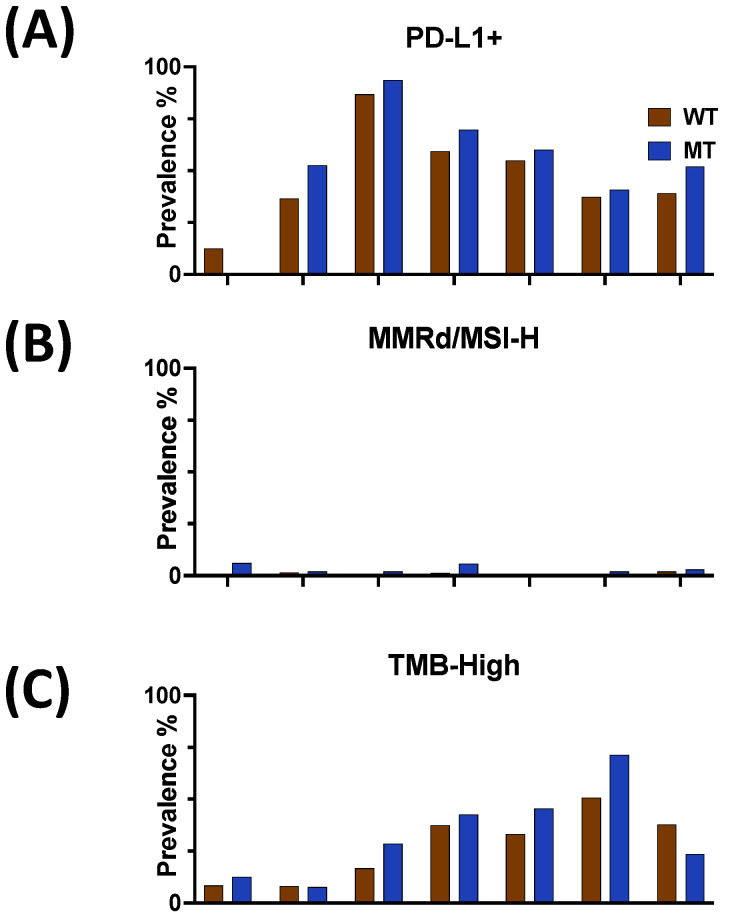
(**A**–**C**) Prevalence of (**A**) PD-L1 IHC (clone 22c3 for all except HN, which used the SP142 clone), (**B**) MMRd/MSI-H, and (**C**) TMB–high positive tumors (comparing MT to WT, no statistically significant differences were observed; *p* > 0.05).

**Figure 4 cancers-16-01572-f004:**
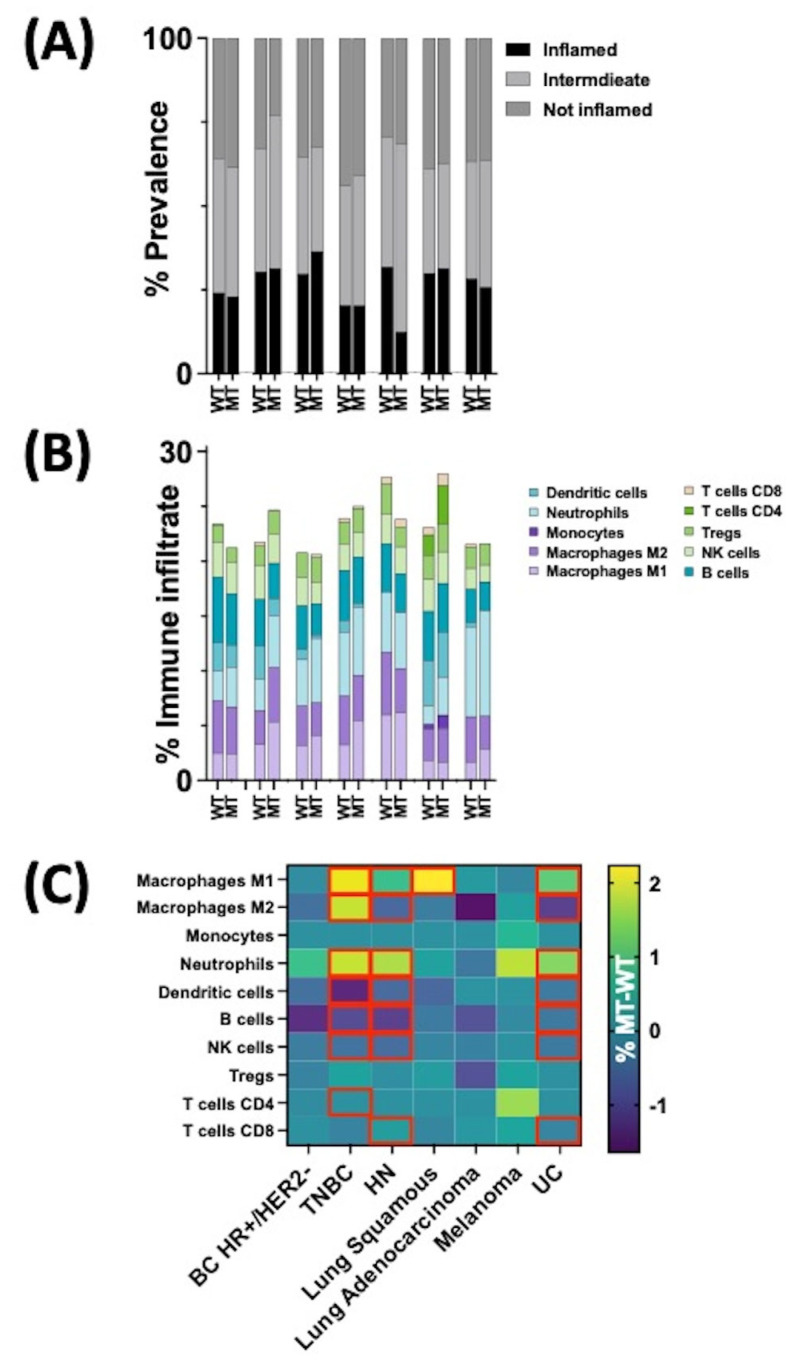
(**A**) Prevalence of T-cell-inflamed tumors across *HRAS* WT and MT tumors (*p* > 0.05, not statistically significant for all). (**B**) Immune cell infiltrates for *HRAS* WT and MT tumors. (**C**) Difference in immune infiltrate percentage between *HRAS* WT and MT tumors (grey box indicates difference is statistically significant (*p* < 0.05).

**Figure 5 cancers-16-01572-f005:**
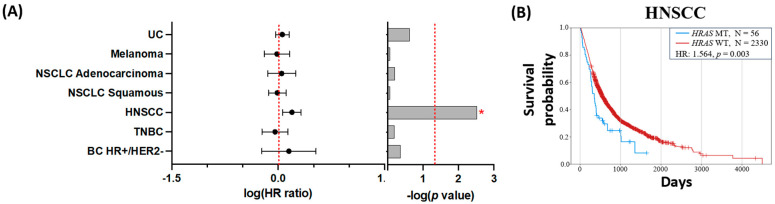
(**A**) Summary of hazard ratio for OS (biopsy to last contact) for investigated solid tumors. (**B**) Kaplan–Meier curve for HNSCC MT vs. WT tumors. A red asterisk [*] indicates statistical significance.

**Table 1 cancers-16-01572-t001:** Demographic information.

UC	*HRAS*mt	*HRAS*wt	*p*-Value
Count (N)	138	4467	NA
Median Age [range]	70 [32–>89]	72 [18–>89]	0.001
Male	66.6% (92/138)	72.2% (3223/4467)	0.158
Female	33.3% (46/138)	27.8% (1244/4467)	
BC			
Count (N)	80	15763	NA
Median Age [range]	68 [41–>89]	60 [19–>89]	0.000
Male	1.3% (1/80)	1.2% (191/15,763)	0.624
Female	98.8% (79/80)	98.8% (15,572/15,763)	
HR+/HER2-	17	8057	<0.001
TNBC	54	4403	
HR+/HER2+	0	688	
HR-/HER2+	0	522	
NSCLC			
Count (N)	150	34,160	NA
Median Age [range]	72 [46–>89] (150)	69 [0–>89] (34,160)	0.004
Male	46% (69/150)	50.0% (17,107/34,160)	0.319
Female	54.0% (81/150)	49.9% (17,053/34,160)	
Adenocarcinoma	22.7% (34/150)	57.1% (19,513/34,160)	0.002
Squamous	60.0% (90/150)	22.0% (7505/34,160)	
Other	17.3% (26/150)	20.9% (7142/34,160)	
Melanoma			
Count (N)	55	5162	NA
Median Age [range]	71 [39–>89] (55)	67 [0–>89] (5162)	0.043
Male	71% (39/55)	62% (3211/5162)	0.231
Female	29.1% (16/55)	37.8% (1951/5162)	
HNSCC			
Count (N)	110	3797	
Median Age [range]	69 [33–>89] (110)	64 [15–>89] (3797)	0.001
Male	62.7% (69/110)	77% (2922/3797)	0.001
Female	37.3% (41/110)	23.0% (875/3797)	

**Table 2 cancers-16-01572-t002:** Summary of hazard ratio (biopsy to last contact) for investigated solid tumors.

	HR	CI	CI	*p*	N *HRAS*WT	N *HRAS*MT
BC HR+/HER2-	1.417	0.589	3.407	0.434	5353	7
TNBC	0.903	0.593	1.374	0.632	3230	35
HN	1.564	1.158	2.11	0.003	2330	56
Lung Squamous	0.971	0.731	1.291	0.842	5893	65
Lung Adenocarcinoma	1.121	0.715	1.759	0.618	15,451	26
Melanoma	0.959	0.636	1.447	0.842	3990	42
UC	1.143	0.917	1.424	0.234	3598	117

## Data Availability

The data used in this study are not publicly available but can be made available to investigators upon reasonable request.

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
