# Peer review of "The Genomic, Transcriptomic, and Immunologic Landscape of *HRAS* Mutations in Solid Tumors"

_cancers, 2024, doi:10.3390/cancers16081572_

Round 1

Reviewer 1 Report

Comments and Suggestions for Authors

The manuscript by Kareff et al attempts to correlate genetic, transcriptomic and immunologic alterations with tumors containing mutant HRAS. The authors use data collected from five different tumor types and correlate genetic alterations, immune signatures, mutational burden, microsatellite instability and clinical outcome with the presence of mutant HRAS. This is a retrospective study with very little positive data. The data presented does not find any significant correlation with the majority of the parameters analyzed with the presence of mutant HRAS. This study is limited by the use of only 5 tumor types and the use of surrogate markers for patient outcomes, as noted by the authors. Furthermore, the number of biomarkers and other genetic alterations associated with mutant HRAS tumors also appears to be somewhat limited. Other than using a larger cohort, this study does not provide any additional information regarding the genetic and immunologic characterization of tumors harboring an HRAS mutation, at least for the 5 tumor types analyzed.

Minor comments:

1) It is unclear from figure 4C to distinguish the differences described in the text due to the color coding/shading in the highlighted areas. 

2) Table A1 may benefit from including the number of samples containing each mutant/variant.

Author Response

Cancers Response to Reviewers (authors’ responses in bold)

The manuscript by Kareff et al attempts to correlate genetic, transcriptomic and immunologic alterations with tumors containing mutant HRAS. The authors use data collected from five different tumor types and correlate genetic alterations, immune signatures, mutational burden, microsatellite instability and clinical outcome with the presence of mutant HRAS. This is a retrospective study with very little positive data. The data presented does not find any significant correlation with the majority of the parameters analyzed with the presence of mutant HRAS. This study is limited by the use of only 5 tumor types and the use of surrogate markers for patient outcomes, as noted by the authors. Furthermore, the number of biomarkers and other genetic alterations associated with mutant HRAS tumors also appears to be somewhat limited. Other than using a larger cohort, this study does not provide any additional information regarding the genetic and immunologic characterization of tumors harboring an HRAS mutation, at least for the 5 tumor types analyzed.

Thank you for the concise summary of our submitted manuscript.

Minor comments:

  • It is unclear from figure 4C to distinguish the differences described in the text due to the color coding/shading in the highlighted areas. 

We have edited the coding and shading in the highlighted areas in Figure 4C on page 12 to improve reader accessibility.

  • Table A1 may benefit from including the number of samples containing each mutant/variant.

We have created a new Table A2 on page 15 including the number of samples containing each mutant/variant in the revised manuscript.

Reviewer 2 Report

Comments and Suggestions for Authors

Minor comments:

1) Please place Figures, Tables and Schemes in the text at the place where they should appear; instead of having them in a separate paragraph.

2) Please cite this work mentioning the involvement of HRAS mutations and increased RAS expression in urothelial carcinoma: J Urol. 2009;181(5):2312-9. doi: 10.1016/j.juro.2009.01.011. Although the prevalence of HRAS mutations was initially though to be higher in urothelial carcinoma, today it is much lower. This can be discussed.

3) In Figure 1A, what is the point of assessing statistical significant differences between different tumors? Which mutation is being assessed here? It is not clear.

4) In Figure 1B, please fix the legend describing WT and MUT tumors. Likewise, the legend in Figure 1C is overlapping the figure showing the expression of HRAS across different HRAS mutations in UC.

5) The legend in Figure 2 depicting the percentage (%) of HRAS MT-WT tumors should be colored (red to blue).

6) In Figure 3B-C, the y-axis should depict percentages with a max value of 100%. Please readjust.

7) Please fix Figure 5B.

8) A few minor typos that need correction:

- Abstract: "...we retrospectively analyzed the tumor tissue of 63,873 tumor tissues..." -> "we retrospectively analyzed 63,873 tumor tissues"

- Results: "...were chosen for a deeper analysis 167 of their molecular and immunological landscaped." -> "...of their molecular and immunological landscapes."

Comments on the Quality of English Language

The English language is good; it just needs some minor modifications.

For example, a few minor typos that need correction:

- Abstract: "...we retrospectively analyzed the tumor tissue of 63,873 tumor tissues..." -> "we retrospectively analyzed 63,873 tumor tissues"

- Results: "...were chosen for a deeper analysis 167 of their molecular and immunological landscaped." -> "...of their molecular and immunological landscapes."

Author Response

Cancers Response to Reviewers (authors’ responses in bold)

Minor comments:

  • Please place Figures, Tables and Schemes in the text at the place where they should appear; instead of having them in a separate paragraph.

We have replaced the Figures and Tables in the Results after first mention or Appendix in the revised submission.

  • Please cite this work mentioning the involvement of HRAS mutations and increased RAS expression in urothelial carcinoma: J Urol. 2009;181(5):2312-9. doi: 10.1016/j.juro.2009.01.011. Although the prevalence of HRAS mutations was initially though to be higher in urothelial carcinoma, today it is much lower. This can be discussed.

Thank you for this suggestion. While the proposal regarding HRAS mutations is interesting, another conclusion from the work regarding mutual exclusivity of HRAS vs. other RAS mutations in the same solid tumors was even more provocative. We edited the text in the on lines 292-295, page 11 below:

“Our results confirm the presence of multiple isoforms of RAS mutations (i.e., KRASmt) within our cohort of HRASmt solid tumors, thereby refuting previous observations that RAS family mutations are mutually exclusive in solid tumors such as urothelial carcinoma.[33]

3) In Figure 1A, what is the point of assessing statistical significant differences between different tumors? Which mutation is being assessed here? It is not clear.

Thank you for highlighting this point. We have added which mutations were assessed into Figure 1A’s text on Page 6 to bolster the take-home message with the below text:

“Q61: 182A>G, 182A>T, 181C>A, 180-182del_insTCT. G12: 34G>A, 34G>T, 35G>A, 35G>T, 35G>C. G13: 37G>C, 38G>T, 38G>A.”

4) In Figure 1B, please fix the legend describing WT and MUT tumors. Likewise, the legend in Figure 1C is overlapping the figure showing the expression of HRAS across different HRAS mutations in UC.

We have fixed the legends of Figures 1B and 1C on Page 6 to improve reader accessibility as recommended.

5) The legend in Figure 2 depicting the percentage (%) of HRAS MT-WT tumors should be colored (red to blue).

We have edited Figure 2’s legend to reflect your recommendation, though we used a blue-yellow color scheme to increase accessibility for readers with achromatopsia, on Page 7.

6) In Figure 3B-C, the y-axis should depict percentages with a max value of 100%. Please readjust.

We have readjusted the y-axis in Figures 3B and 3C as recommended on Page 8.

7) Please fix Figure 5B.

We have also fixed Figure 5B as recommended on Page 10.

8) A few minor typos that need correction:

- Abstract: "...we retrospectively analyzed the tumor tissue of 63,873 tumor tissues..." -> "we retrospectively analyzed 63,873 tumor tissues"

Thank you for highlighting. We removed the first instance of “the tumor tissue of” on page 1, line 20.

- Results: "...were chosen for a deeper analysis 167 of their molecular and immunological landscaped." -> "...of their molecular and immunological landscapes."

Thank you for catching. “Landscapes” is spelled correctly on page 4, line 163.

Round 2

Reviewer 1 Report

Comments and Suggestions for Authors

The authors have modified the manuscript appropriately.